# High-throughput method characterizes hundreds of previously unknown antibiotic resistance mutations

Matthew J. Jago [1], Jake K. Soley [1,2], Stepan Denisov [1], Calum J. Walsh[2], Danna R. Gifford [1], Benjamin P. Howden [2,3] & Mato Lagator [1]✉

A fundamental obstacle to tackling the antimicrobial resistance crisis is identifying mutations that lead to resistance in a given genomic background and environment. We present a high-throughput technique – Quantitative Mutational Scan sequencing (QMS-seq) – that enables quantitative comparison of which genes are under antibiotic selection and captures how genetic background influences resistance evolution. We compare four *E. coli* strains exposed to ciprofloxacin, cycloserine, or nitrofurantoin and identify 812 resistance mutations, many in genes and regulatory regions not previously associated with resistance. We find that multi-drug and antibiotic-specific resistance are acquired through categorically different types of mutations, and that minor genotypic differences significantly influence evolutionary routes to resistance. By quantifying mutation frequency with single base pair resolution, QMS-seq informs about the underlying mechanisms of resistance and identifies mutational hotspots within genes. Our method provides a way to rapidly screen for resistance mutations while assessing the impact of multiple confounding factors.

Antimicrobial resistance (AMR) is an ongoing global health crisis limiting our capacity to treat infections. AMR infections lead to five million deaths each year, and this burden is predicted to increase considerably[1]. AMR often arises through chromosomal mutations that alter susceptibility to one or more antibiotics[2]. Consequently, predicting drug susceptibility from genomic data is becoming vital to antimicrobial stewardship by informing and stratifying treatments[3]. However, the accuracy of such predictions is limited by our existing knowledge of which mutations contribute to AMR and through what mechanisms.

Mapping mutations to AMR phenotypes is typically achieved through sequencing, either of clinical isolates or laboratory strains that have undergone selection for resistance. These approaches have proven powerful tools but are often biased towards mutations that confer high-level resistance and struggle to discern how AMR evolution varies in different contexts. Smaller-effect mutations, epistatic interactions with genetic background, and environmental factors contribute significantly to overall resistance levels, complicating genomic predictions[4,5]. As such, a key unmet challenge is to quantify the potential mutation space for resistance beyond the well-trodden evolutionary paths we currently understand.

To achieve this, we developed Quantitative Mutational Scan sequencing (QMS-seq), a mutant screening technique that adapts metagenomic sequencing to quickly characterize mutational landscapes for resistance. In QMS-seq, a genetically homogeneous population is allowed to accumulate random mutants over 24 h under minimal selection—growing in rich media without antibiotics—to produce a heterogeneous population where most variants contain a single

[1]Division of Evolution, Infection and Genomic Sciences, School of Biological Sciences, Faculty of Biology, Medicine and Health, University of Manchester, Manchester M13 9PL, UK. [2]Department of Microbiology and Immunology, University of Melbourne, at the Peter Doherty Institute for Infection and Immunity, Melbourne, VIC 3000, Australia. [3]Centre for Pathogen Genomics, University of Melbourne, Melbourne, VIC 3000, Australia. ✉e-mail: mato.lagator@manchester.ac.uk

mutation (Fig. 1A). This population is spread across ten selective agar plates, here containing the minimum inhibitory concentration (MIC) of an antibiotic. After resistant colonies have grown, they are mixed and sequenced collectively with sufficient depth to detect low-frequency resistance mutations. A custom bioinformatic pipeline (Fig. 1B) stringently filters for high-quality reads and utilizes the high specificity/ sensitivity software *lofreq*[6] to call single-nucleotide variants and small indels. Another tool, *breseq*[7], identifies larger mobilization events for known insertion sequences. QMS-seq captures a comprehensive landscape of mutations under selection in a given condition, enabling combinatorial comparison across different environments and genetic backgrounds.

Here, we employ QMS-seq to classify which mutations provide resistance to a single versus multiple antibiotics and to investigate how carrying resistance to one antibiotic influences the evolution of resistance to another. We take three different strains of rifampicin-resistant *Escherichia coli* and evaluate their mutational landscapes for three secondary antibiotics (Fig. 1C). Each strain carries a different non-synonymous mutation in *rpoB*, the β subunit of RNA polymerase. Besides providing resistance to rifampicin, these mutations also cause substantial variation in gene expression between the strains[8]. By using secondary antibiotics that target different cellular pathways (ciprofloxacin/CIP: DNA replication[9], cycloserine/CYC: cell wall biogenesis[10],

nitrofurantoin/NIT: protein synthesis[11]), we quantify how antibiotic mechanism and genetic background cooperatively shape the identity of resistance mutations. We also identify hundreds of mutations not previously associated with resistance, reveal fundamental differences between mutations providing multi-drug versus antibiotic-specific resistance, and provide insight into the mechanisms underpinning resistance.

## Results
### Mutational landscape for resistance
We identified 812 mutations across all conditions, in 251 genes and 49 regulatory features (Fig. 2A and Supplementary Data 1–2). We consider these resistance mutations because they emerged under strong selection in colonies that were able to grow at 1x MIC. Conservative filtering criteria verified for strong positive selection and excluded 60% of the mutations originally called by our pipeline (see *Methods*). This ensured that each of the 812 mutations is likely the underlying genetic cause of the resistance phenotype, i.e., growth at elevated antibiotic concentrations (for more, see *Discussion*). We quantified every mutation by its "occurrence": the number of independent samples (out of 35) that it was identified in. Interestingly, many more mutations occurred within intergenic regions (37%) than expected given the composition of *E. coli*'s genome (13% intergenic), suggesting that

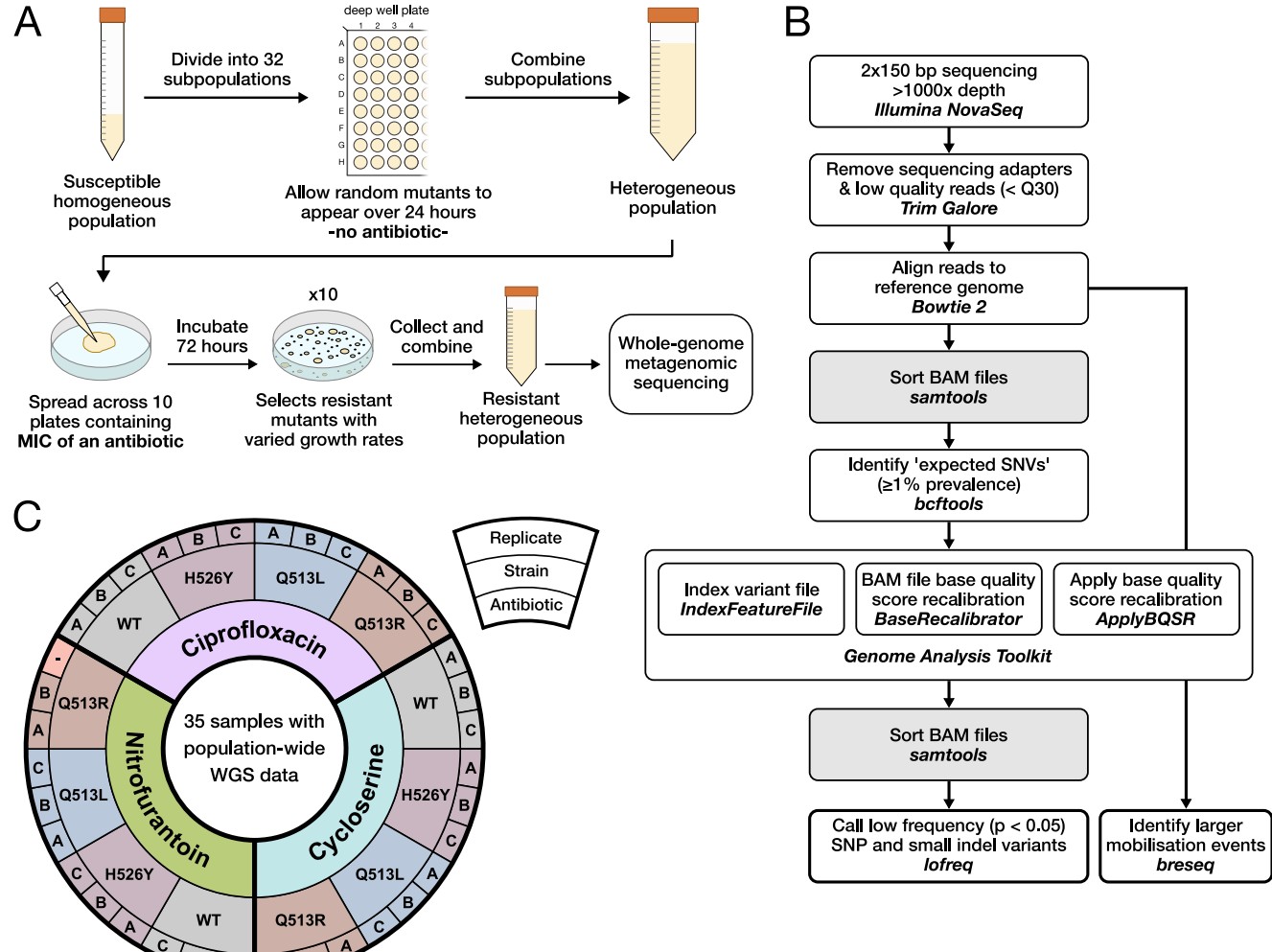

**Fig. 1 | Quantitative Mutational Scan sequencing. A** Experimental technique. The 24-h mutant accumulation step uses 32 subpopulations to reduce the final population's bias towards mutants that by chance arise earlier. **B** Sequencing and variant calling pipeline. **C** Experimental design. We tested three antibiotics and four strains of *E. coli*, performing three replicates for each of the 12 conditions, comprising 36 independent samples. The wild type (WT) strain is BW25113, which was the ancestor of the three rifampicin-resistant strains with mutations in *rpoB*. One nitrofurantoin/ Q513R replicate was excluded from the analysis due to sequencing failure.

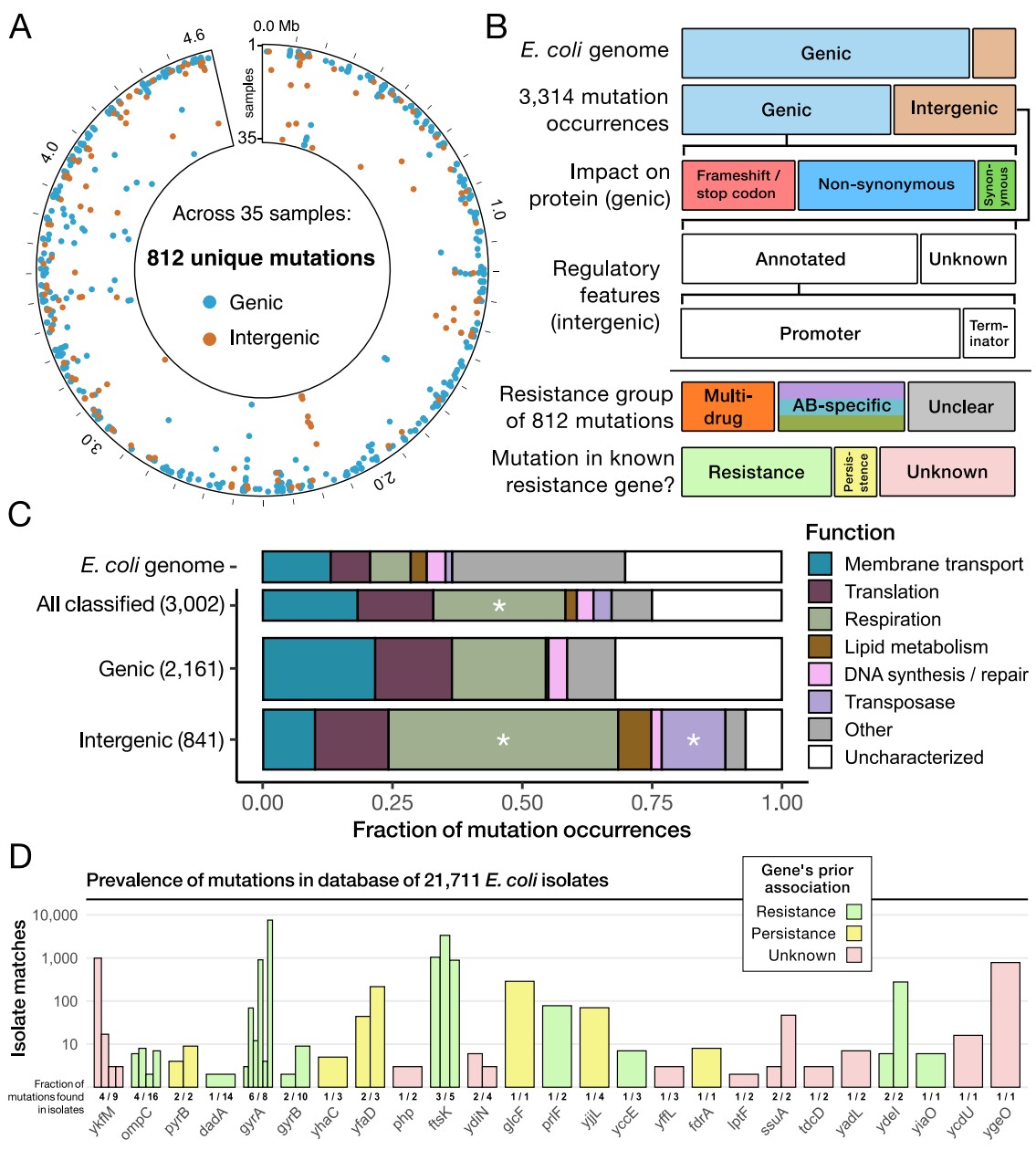

**Fig. 2 | Summary of resistance mutations identified by QMS-seq.**
**A** Circumferential axis is the mutation's location in the *E. coli* genome (4.6 × 10⁶ bp). Radial axis is the mutation occurrence: the number of samples each mutation was identified in. Color indicates if the mutation was located within a gene or an intergenic sequence. **B** Genic mutations were further categorized based on how they affect the amino acid sequence; intergenic mutations based on whether they appear in known regulatory features[59,60]. Mutations were considered "multi-drug" if they were found in samples from all three antibiotics, and "antibiotic-specific" if they were significantly over-represented in one (see *Methods: Categorizing antibiotic-specific and strain-specific targets*). For every gene, any previous association with antibiotic resistance or persistence was confirmed through a literature search. **C** All genic and annotated intergenic mutation occurrences were categorized by the biological function of the gene they affect. Also shown is the same classification of the WT genome (4487 genes). Stars indicate categories that were significant (0.01 < P < 0.05) in gene set enrichment analysis. **D** For every mutation, we checked if it was also present in any genomes from a WGS database for global human/animal-sourced *E. coli* isolates[18]. Each bar represents one mutation, its height indicating the number of isolates with a matching sequence. Shown are the 25 genes with the most matches, ordered by their mutation occurrence in our samples and colored according to any previous association with antibiotic resistance or persistence. Source data are provided as a Source Data file.

regulatory changes contribute to resistance evolution more than typically thought[12] (Fig. 2B). Nearly half the mutations were in genes that have not previously been associated with resistance to any antibiotic, including 11 of the 30 most targeted genes. This indicates how much of the mutational landscape for resistance remains uncharacterized and emphasizes QMS-seq's utility in exploring it. Surprisingly, 13% of mutations affected genes known to be involved in antibiotic persistence—a transient tolerance phenotype that arises

through the dysregulation of such genes[13]. The observation of heritable mutations in persistence genes suggests overlapping mechanisms for resistance and persistence at MIC.

Most of the targeted genes with known functions are involved in either membrane transport, translation, or cellular respiration (Fig. 2C), which are all associated with antibiotic resistance[14–17]. The considerable variety of genes in each category suggests that altering membrane architecture or core cellular processes is a frequent

strategy for acquiring MIC-level resistance. There were notable differences between genic and regulatory mutations—the former much more common in membrane transporters, the latter over-represented in genes associated with respiration. Additionally, we found mutations in multiple promoters for lipid metabolism genes and different copies of the *insH1* transposase, without equivalent genic mutations. Despite substantial overlap of functional categories between genic and regulatory mutations, the disparities indicate which resistance mechanisms favor differential expression over protein modification.

To verify the mutations are also relevant to non-laboratory settings, we searched for matches in 21,711 genomes of clinical and environmental *E. coli* isolates from the BV-BRC database[18] (Fig. 2D). There were 50 genes (of 251 we identified) whose mutation(s) matched the sequence of one or more isolates, including several well-characterized genes associated with clinical resistance. Six of the eight mutations we identified in *gyrA* (encodes CIP's primary target) were present across hundreds of isolates, while the ten *gyrB* (secondary target) mutations were much less prevalent. Mutations in *ompC* were also represented, a membrane porin sometimes mutated in clinical *E. coli* with multi-drug resistance[19]. Interestingly, there were hundreds of matches for *ykfM*, an uncharacterized membrane protein with the highest mutation occurrence across all three antibiotic conditions. The mutations clustered at the boundary between *ykfM*'s predicted transmembrane helix and cytoplasmic domains (Supplementary Fig. 1), suggesting it may also encode a porin that can be modified to reduce intracellular antibiotic levels. The presence of previously unknown resistance mutations from our study in clinical isolates demonstrates QMS-seq's ability to accelerate the mapping of genomic information to resistance phenotypes.

### Multi-drug versus antibiotic-specific resistance

Our experimental design (Fig. 1C) can differentiate between multi-drug resistance (MDR) mutations that confer resistance to all three antibiotics and mutations that confer antibiotic-specific resistance (ASR). We found categorical differences in both their intragenic positioning and their typical impact on the encoded protein. MDR mutations almost exclusively clustered within a small region (<15%) of the gene, even for genes with many unique mutations (Wilcoxon signed-rank test: $W = 155$, $P < 0.001$) (Fig. 3A). Furthermore, mutations leading to MDR were less likely to be "high-impact" (nonsense or frameshift) while "moderate-impact" (non-synonymous or in-frame) and "low-impact" (synonymous) mutations were significantly more frequent in MDR ($\chi_8 = 428$; $P < 0.001$) (Fig. 3B). Conversely, for most of the top ASR targets, we identified high-impact mutations across the entire length of the gene, indicating loss-of-function provided resistance. Supporting this conclusion, over 75% of the insertion element mobilizations we detected happened within ASR genes (Supplementary Data 3). MDR mutations occurred more frequently than ASR mutations and accounted for 89% of intergenic mutations (Fig. 3C), implying that expression-mediated resistance is less dependent on the antibiotic's mechanism of action. Therefore, at MIC, the most common mutations are active site modifications and regulatory changes that provide MDR, whereas ASR typically arises from knockout mutations.

Genes targeted for MDR were spread across a wide range of functional categories, without significant bias towards one (Fig. 3D). In contrast, ASR mutations were more likely to target specific functions ($\chi_{28} = 983$; $P < 0.001$), with translation and DNA synthesis/repair favored by CIP, respiration by CYC, and membrane transport by NIT. Given that CIP causes DNA damage and inhibits gyrases, which are active during DNA synthesis and (co-transcriptional) translation, the antibiotic mechanism of action may explain the observed biases in targeted functional categories. As such, the evolutionary pathways for ASR reflect the antibiotic's mechanism of action. Among the top ASR genes were well-documented targets of resistance mutations: *gyrA* and *icd* for CIP[20,21], *ispA/B* and *ubiA/E/G/H* for CYC[22,23], and *ompC* for NIT[19].

*ompC* is unique as it is typically considered a target for MDR, rather than being NIT-specific—yet despite mutations in every NIT sample, we identified none in CIP or CYC. Interestingly, with CIP's secondary target (*gyrB*), we saw a cluster of CIP-specific mutations in the described quinolone-resistance region, but also several mutations outside this region in CYC and NIT (Supplementary Fig. 2), pointing to a role for *gyrB* in MDR. After recreating mutant strains for nine of the most common MDR and ASR gene targets, we confirmed all but one were resistant to at least 1x MIC of the antibiotic(s) they were initially selected in (see *Methods: validating resistance genes*).

### Genotype influences resistance evolution

We utilized four almost identical *E. coli* strains: the WT ancestor and three descendants, each with a single substitution in RNA polymerase that provides resistance to rifampicin and alters their transcriptome[8] (Fig. 1C). We quantified the differences in mutational landscapes associated with each strain to evaluate how varying one point mutation in the genomic background influences the evolution of resistance to secondary antibiotics. We observed a clear interaction between CIP and the Q513R strain—the number of unique genes targeted in this condition was significantly higher than any other ($F_{3,8} = 6.2$; $P < 0.05$) (Fig. 4A). CIP/Q513R mutation count correspondingly high, on average nearly double the other conditions, which otherwise remained relatively consistent between antibiotics and strains (Supplementary Fig. 3). Over 60% of the mutations unique to CIP/Q513R were insertions or deletions, compared to 24% globally, suggesting faulty DNA repair played a role in the effect. In support of this, we identified several mutations exclusive to CIP/Q513R samples in critical genes for two DNA repair pathways, *recO*[24] and *uvrB*[25]. Principal component analysis of the twelve conditions (Fig. 4B) indicates the heavy influence this effect had on the mutational landscapes, as CIP/Q513R is responsible for most of the variance. NIT/Q513L also separates from the other conditions, although this comes solely from having *different* genetic targets, rather than more genes being targeted as in CIP/Q513R. The other conditions broadly cluster by antibiotic, while the small distance between antibiotic groups reflects the prevalence of MDR mutations (Fig. 3C).

At the individual gene level, strain-specific effects were more common in genes that were also specific to an antibiotic, although the mutational distributions of several MDR genes were shaped by an interaction *between* the antibiotics and strains (Fig. 4C). These distributions also show the influence of CIP/Q513R and NIT/Q513L. One gene (*stfP*) was targeted more often in both these conditions, while *xylB* and *ycgX* were absent despite being targeted in most other conditions. Many antibiotic-specific genes were unique to CIP/Q513R and NIT/Q513L, helping explain their positions along the principal components. However, less substantial interactions were also revealed for other strains, including the absence of many CIP targets in H526Y and CYC targets in Q513R. Our data demonstrates how a difference of just one nucleotide between strains can result in distinct evolutionary trajectories under antibiotic selection. While the exact relationship between the transcriptomic effects of rifampicin resistance mutations and the evolution of secondary resistance remains of interest, QMS-seq provides a way to evaluate these complicated interactions between antibiotics and genetic background.

### QMS-seq provides insight into the mechanisms of resistance

Studies of resistance typically focus on mutations that change the amino acid sequence, as their effects are well-characterized and they are generally considered more important than synonymous or non-coding mutations[26]. However, growing evidence suggests mutations that impact gene expression levels can also be phenotypically significant[27,28] and clinically relevant[29–31]. The following case studies explore several unique examples of mutations that do not alter a polypeptide but were still under strong antibiotic selection, and what they may suggest about less conventional resistance strategies.

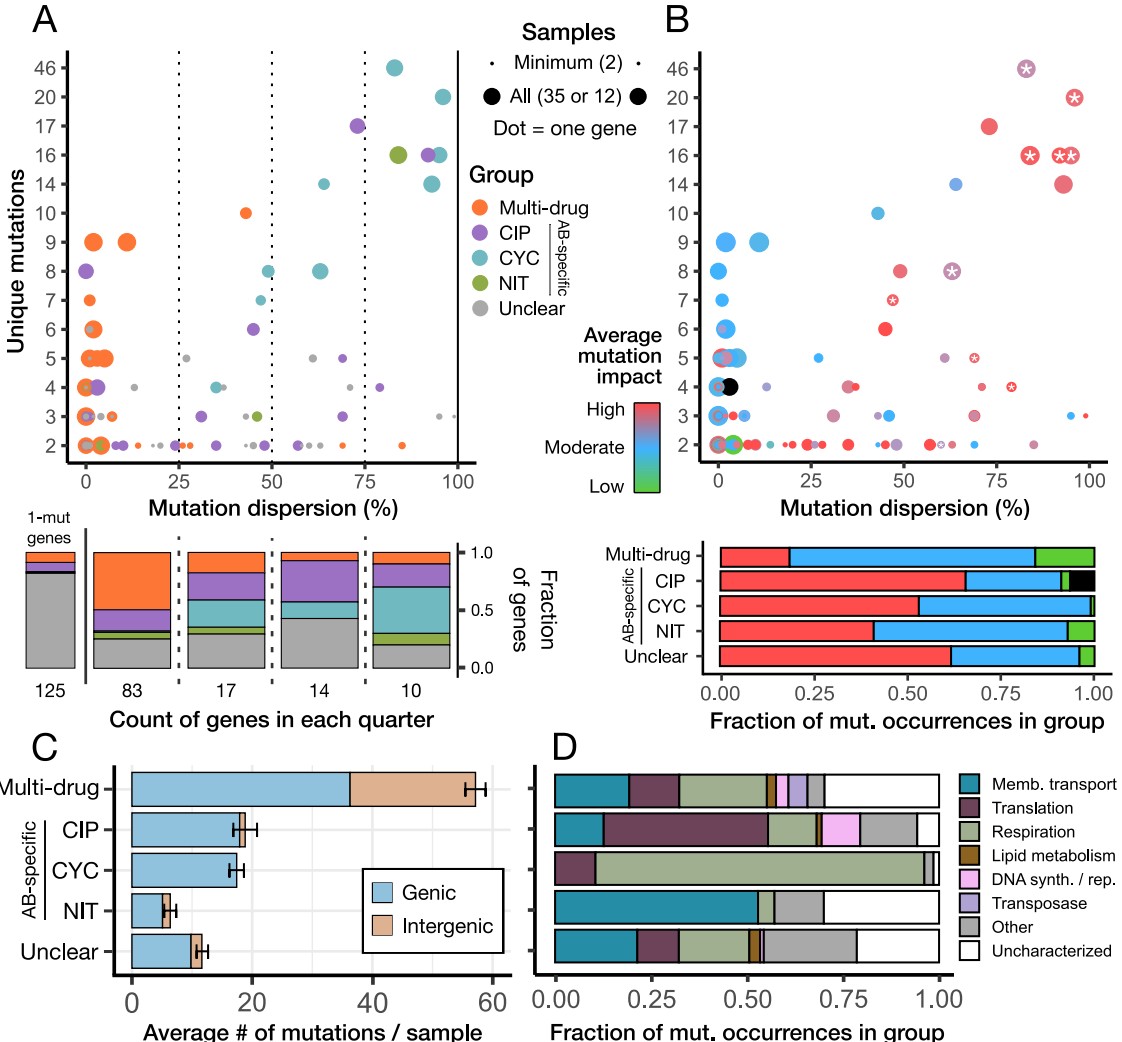

**Fig. 3 | Multi-drug versus antibiotic-specific resistance mutations. A** Every gene arranged by the number of unique mutations identified in that gene and how much the mutations dispersed throughout the gene. Dispersion was defined as the mean distance between adjacent mutations, divided by the maximum possible distance if all mutations were equally far apart. In other words, low dispersion means that mutations clustered in a specific region of the gene, while higher values indicate mutations were spread across the entire length of the gene. The color shows if the gene was classified as multi-drug, antibiotic-specific, or unclear (those with insufficient mutation occurrences to categorize confidently). Size indicates the proportion of samples where mutations in that gene were observed, out of a maximum of 12 for antibiotic-specific genes or 35 for multi-drug and unclear genes. The bars below summarize each quarter of the plot, with an additional bar on the left representing genes that cannot be displayed on the plot's axes, as only one unique mutation was observed. **B** The same plot, with points colored by the average impact type of all mutation occurrences in that gene. High: nonsense and frameshift, Moderate: non-synonymous and in-frame, Low: synonymous. Black indicates a non-coding RNA. Stars indicate that we also detected at least one occurrence of an insertion element mobilization event within the gene. Below is the impact-type distribution of all mutations occurring in MDR vs. ASR genes. **C** The mean number of mutations associated with MDR vs. ASR in any one sample. Error bars are SEM (multi-drug and unclear: *n* = 35, CIP and CYC: *n* = 12, NIT: *n* = 11). **D** Proportion of MDR vs. ASR mutations associated with different functional categories. Source data are provided as a Source Data file.

**Regulating expression with promoters, terminators, and start codons.** The over-abundance of intergenic mutations (Fig. 2B) makes it clear they are major contributors to resistance at 1x MIC. To illustrate some mechanisms through which regulatory changes might have been achieved, we focus on three genes whose annotated promoter or terminator sequences were targeted by high-occurrence mutations (Fig. 5A). There were several mutations in the promoter for *yheO*, a transcription factor involved in responding to multiple stressors[32] and predicted to regulate tRNA modification genes[33]. Change in the expression of a global regulator like *yheO* could reverberate through the regulatory network, potentially affecting the expression of many other genes. Two mutations, one of which was present in every sample, were found in the terminator of *lpxC*, an LPS biosynthesis gene. While genic mutations in *lpxC* can modify antibiotic susceptibility[34], terminator mutations are more likely to influence the expression of the downstream operon[35]. This encodes three genes, including *mutT*, which is responsible for removing oxidized guanine from the nucleotide pool[36]. Guanine oxidation is an underlying mechanism of several bactericidal antibiotics, and overproduction of MutT protects *E. coli* from antibiotic killing[37]. If the *lpxC* mutations increase terminator read-through, they could elevate *mutT* expression to provide resistance. A unique mutational target was the uncharacterized lipoprotein *yiiG*, where we observed two mutations in the promoter but also a mutation in its start codon. The latter converted ATG (89% of *E. coli* start codons) into ATT (0.14%), potentially lowering *yiiG* expression by limiting translation initiation[38]. As such, *yiiG* points to different mutational strategies achieving the same outcome—alteration of gene expression level.

**Isoleucine codons as an evolutionary toggle switch.** Synonymous mutations made up 11% of the dataset, with several present in nearly

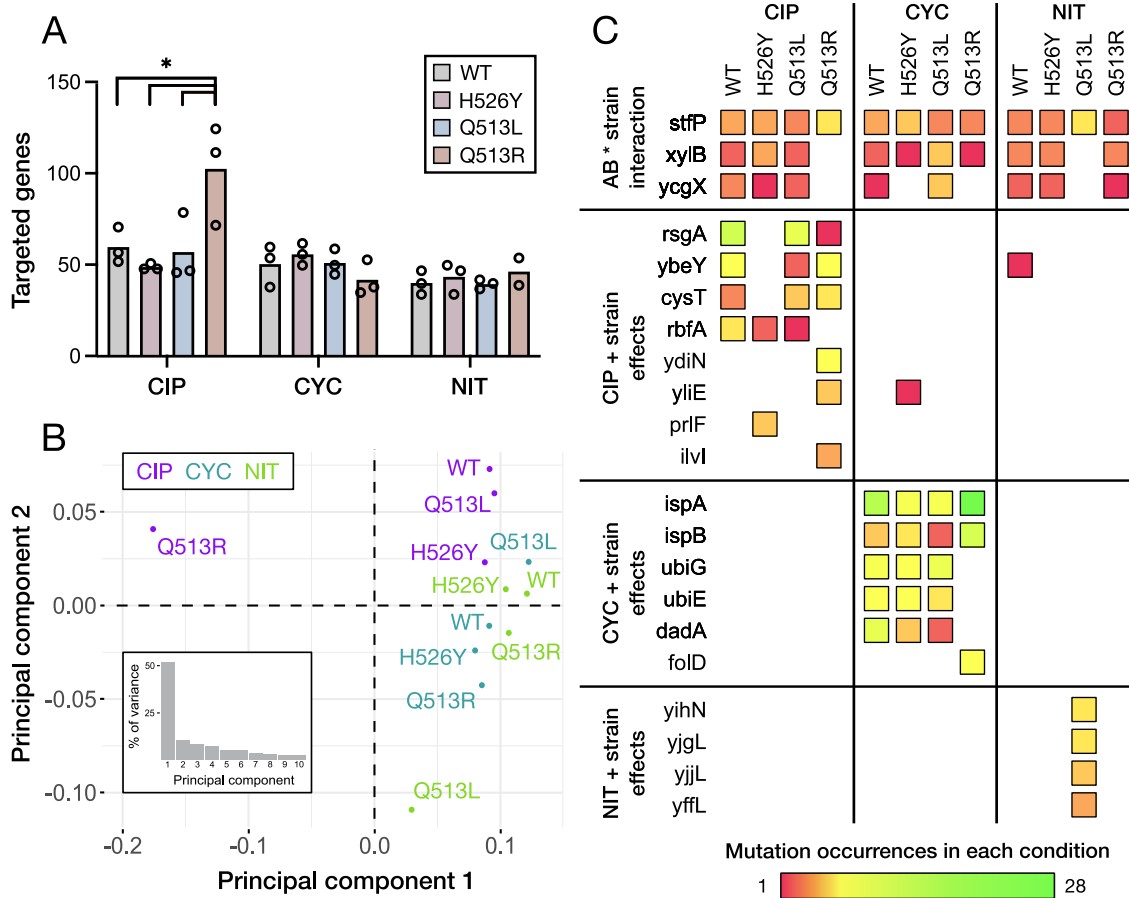

**Fig. 4 | Influence of genetic background on resistance evolution. A** The mean number of targeted genes (including regulatory targets) in samples from each of the 12 different conditions, with individual replicates shown ($n = 3$). The star indicates significance in a one-way ANOVA ($P = 0.021$) with FDR correction for multiple comparisons. **B** Principal component analysis of the 12 antibiotic/strain combinations, using the number of samples (0–3) from each condition that every mutation was observed in. The inset panel is a scree plot indicating the percentage of variance described by each principal component. **C** The likelihood of a gene being targeted by mutations can depend on the strain's genetic background. This was observed for MDR genes (top row) as well as ASR genes (bottom three rows). Shown are the top mutational targets with significant strain effects, determined by a generalized linear model. Source data are provided as a Source Data file.

every sample, evidencing their importance to all three antibiotics. Incredibly, four of the five most common synonymous mutations were identical, swapping an Ile-ATC codon for Ile-ATA. The *rplF* and *pyrB* genes demonstrate just how specific these mutations were, as nearby Ile codons that are not ATC were untouched, as were other Ile-ATC codons further away in the gene (Fig. 5B). ATA is one of the least common codons in *E. coli* and has a unique decoding mechanism that requires a post-transcriptionally modified tRNA[39]. Rare codons like ATA take longer to decode, which can slow translation at their locus through ribosomal pausing[40]. When near a gene's start (like in *rplF* and *pyrB*), synonymous mutations can also impact the secondary structure of mRNA, affecting the rate at which ribosomes bind to initiate translation[41]. The synonymous Ile-ATA mutation in *htrL* adds an intriguing twist, as it was located in a cluster of non-synonymous mutations —indicating the mechanism may be related to the protein's fold. Synonymous mutations' influence on translation rate can also affect protein structure since polypeptides begin to fold during translation[42]. By occurrence, nearly two-thirds of all synonymous mutations were ATC > ATA, despite ATC codons making up only 2.5% of the *E. coli* genome, which points to a conserved and previously unappreciated mechanism of adaptation.

**Should-be-lethal tRNA mutations provide ciprofloxacin resistance.** We observed CIP-specific mutations in the *cysT* tRNA that altered its 3' CCA tail, a conserved feature among tRNAs that is required for aminoacylation[43] (Fig. 5C). Although unprecedented, this is not entirely without basis; co-treatment of CIP with cysteine (the amino acid for *cysT*) was shown to significantly increase killing of persister cells[44]. The synergistic relationship between CIP and cysteine appears to be recapitulated in our experiments, except through heritable mutations. However, *cysT* is the only cysteine tRNA in the genome, posing a puzzle as to how disruptions of its 3' CCA are not lethal. *E. coli* carries a 3' CCA repair mechanism[45], which is likely able to rescue enough cysteine tRNA molecules to prevent death, presumably resulting in a variant with reduced translation of nearly every protein. That this strategy is unique to the *cysT* tRNA may suggest the resistance mechanism is dependent on global regulatory shifts associated with the frequency of cysteine residues across different proteins. Furthermore, *cysT* is a prominent example of evolutionary differences between strains: all CIP samples contained at least one *cysT* mutation, except the three H526Y samples, which had none (Fig. 4C). Determining how *cysT* plays into the synergistic effect of CIP and cysteine will require further experimentation, but our findings highlight the potential avenues for research opened by QMS-seq's ability to identify rare and low-fitness resistance mutations.

## Discussion

We used QMS-seq to characterize the mutational landscapes for resistance to three different antibiotics, identifying 812 mutations across 300 coding and regulatory sequences. The re-occurrence of

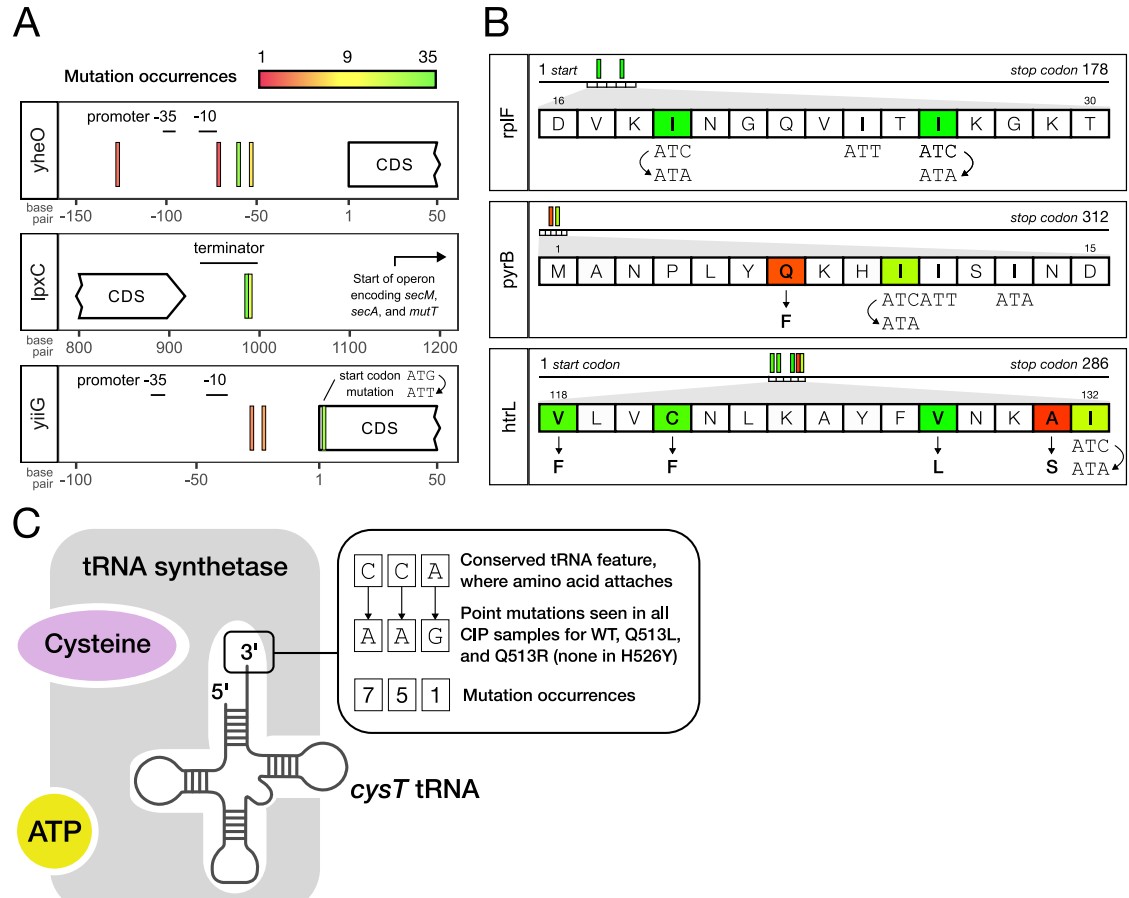

**Fig. 5 | Case studies of notable resistance mutation targets. A** Three genes that were targeted by high-occurrence mutations in their annotated regulatory sequences[59,60], including one with a start codon mutation. **B** Genes with the highest occurrence of synonymous mutations. These mutations were all in the same codon, swapping Ile-ATC to Ile-ATA, even though nearby Ile-ATT or -ATA codons were not targeted. Some of the genes were also targeted by non-synonymous mutations (downward arrows). **C** Cysteine tRNA aminoacylation (where *cysT* is charged with the cysteine amino acid by tRNA synthetase). We observed ciprofloxacin-specific mutations in *cysT*, which, in principle, should prevent its interaction with cysteine.

mutations across independent samples provided a quantitative metric that describes how mutational effects depend on experimental conditions, helping evaluate the genetic and environmental factors that shape resistance evolution. Through this, QMS-seq contributed valuable insights into microbe biology and shed light on the critical question of how resistance to one antibiotic influences the evolution of resistance to another[46].

QMS-seq enabled the discovery of many genes and regulatory sequences with previously unknown roles in resistance. This was facilitated by three design choices: (i) Using parallel subcultures during mutant accumulation, which stops early-occurring mutants from dominating the final population[47] to help capture a broader mutational landscape. (ii) Setting minimum inhibitory concentration (MIC) as the selective threshold, since most clinical isolate and experimental evolution studies identify mutations from populations exposed to much higher antibiotic concentrations. (iii) Focusing on mutation occurrence, by considering how often a mutation appears across independent samples instead of how prevalent it is within a population. Occurrence quantifies the likelihood of each mutation in the chosen genetic background, given that the mutation is under sufficient selection in the chosen antibiotic condition. It largely ignores fitness costs of mutations, and the level of resistance conferred above the selective threshold, explaining why we identified so many previously unknown resistance targets. This is noteworthy, as mutations selected at lower antibiotic concentrations are often clinically significant[30], and high-level resistance can evolve incrementally through the

accumulation of smaller-effect mutations[4] or epistatic interactions magnifying their individual effects[5].

Using the MIC may also have contributed to the surprising abundance of regulatory and synonymous mutations, sites often thought to be under weak selection[12]. Their frequency here corroborates the notion that such mutations can be an important part of resistance evolution[29–31]. Non-coding mutations may be enriched during selection at MIC because they are less likely to have a high-fitness cost or be lethal[48], and the effect they have on gene expression is sufficient to mediate low-level resistance. Regardless, the case studies emphasize how much remains unknown about the mechanisms of regulatory mutations and their role in antibiotic resistance. Regulation is wholly responsible for antibiotic *persistence*, where temporary changes in gene expression induce a slow- or no-growth state, protecting a small fraction of the population from antibiotic killing[13]. The mutations we observed in persistence genes may provide a similar slow-growing tolerance phenotype that would be permanent and heritable. The clearest evidence for this is the *cysT* mutations (Fig. 5C), which most likely slow growth and are linked to cysteine's ability to potentiate the killing of persister cells[44].

A key strength of QMS-seq is simultaneously identifying hundreds of mutations throughout the genome, which comes from pooling mutant colonies and sequencing collectively. An inherent consequence is that it's impossible to determine if further mutations have arisen after an initial resistance mutation, in the same lineage. Our filtering criteria (see *Methods*) use re-occurrence across samples to

verify for strong selective pressure and should exclude hitchhiker mutations. However, they would not exclude subsequent mutations that were under strong positive selection in the given resistance background, such as compensatory mutations. A clear example is CIP/Q513R, where DNA repair mutations only found in those samples significantly increased the number of mutations identified (Fig. 4A). A hypermutator arising during mutant accumulation would facilitate the acquisition of additional mutations to provide resistance[49]. Most non-resistance mutations identified from these variants would be filtered out, but if the strategy was particularly effective, the initial DNA repair mutation would also be found in multiple samples and pass our filtering. Thus, despite filtering out 60% of mutations identified by the variant calling pipeline, a small number of mutations in our final dataset may not directly provide resistance. Individually sequenced colonies largely had only one mutation, none more than two (Supplementary Table 1), suggesting a vast majority of the final mutations are the underlying cause of the resistance phenotype. Ultimately, QMS-seq serves to comprehensively identify all mutations that contribute to resistance, which can include compensatory mutations under strong selection to independently emerge in multiple replicates.

QMS-seq is experimentally simple, relatively inexpensive, and capable of rapidly identifying resistance mutations across an entire genome while distinguishing between those providing multi-drug versus antibiotic-specific resistance. It affords sufficient resolution to build a map of which sequence features in those genes are targeted and how—helping to uncover the underlying mechanisms of resistance. Importantly, QMS-seq evaluates the above in a strain-specific context, enabling researchers to disentangle how resistance evolution depends on the genotype of strains under investigation. In doing so, our method addresses key deficiencies in the effort to monitor resistance evolution from genomic data and can be employed to investigate questions critical to antimicrobial stewardship, as well as provide a deeper view into the nature of microbial evolution. QMS-seq's broad applicability could help transition from a largely reactive response to the AMR crisis, towards pre-emptive strategies informed by studies examining diverse strains' evolvability under a variety of conditions.

## Methods

### Experimental strains
The wild type (WT) strain was Escherichia *coli* BW25113 (NZ_CP009273.1) from the Keio collection[50] containing a knock-out of the *tolC* gene. The three *rpoB* mutants, which are rifampicin-resistant, evolved from this ancestor[8]. Whole-genome sequencing confirmed that the strains differ by only one point mutation in *rpoB*, each causing a different amino acid substitution: H526Y, Q513L, or Q513R. The ΔtolC strain was chosen so that only *rpoB* mutations were selected for by rifampicin, since TolC is a prominent efflux channel and a common target for antibiotic resistance mutations. Its absence from the strains in this study may have increased the relative frequency of alternate resistance mutations and further contributed to the observed diversity of resistance mutation targets.

### Antibiotic selection and MIC determination
We chose three antibiotics with distinct mechanisms of action to demonstrate QMS-seq's broad applicability, and to identify multi-drug resistance mutations that provide resistance to multiple different mechanisms of action. Ciprofloxacin targets DNA topoisomerase II (gyrase), which causes replication fork blockage and DNA damage, inhibiting growth before eventually killing the cells[9]. Cycloserine inhibits growth by interfering with cell wall biosynthesis: as a structural analog of D-alanine, it competes to bind alanine racemase and D-alanine—D-alanine ligase, disrupting peptidoglycan production[10]. Nitrofurantoin is reduced into various metabolites, some of which are highly reactive and non-specifically target ribosomes to prevent protein synthesis[11].

We defined minimum inhibitory concentration (MIC) as the lowest concentration of a given antibiotic that prevented colonies from forming on an agar plate. We determined the MIC by plating 50 µL of overnight culture on a series of Mueller Hinton (MH) agar plates supplemented with two-fold serial dilutions of an antibiotic, then incubating the plates for 24 h at 37 °C. We tested the WT and all three *rpoB* mutants separately, but no differences in MIC were identified between strains. The MIC was 0.005 µg/mL for ciprofloxacin (Sigma-Aldrich), 100 µg/mL for cycloserine (Sigma-Aldrich), and 10 µg/mL for nitrofurantoin (TOKU-E).

### Selecting for antibiotic resistance mutations
We inoculated MH media with glycerol stock of an experimental strain, then immediately split the inoculation into three tubes to start independent replicate populations. After growing the three cultures overnight at 37 °C, we diluted them 1:10,000 in MH and split each into 32 subpopulations across a deep 96-well plate. We incubated the cultures for 24 h at 37 °C, allowing random mutations to occur during normal growth under minimal selective pressure. The $3 \times 32$ subpopulations were then pooled into three independent heterogeneous populations (triplicates) and spread on MH agar plates containing the MIC of an antibiotic to select for resistance mutations.

This design was inspired by Luria-Delbrück experiments, which use parallel subcultures to improve estimates of mutation rate from counts of resistant clones in a population[47]. Clones from a mutant that arises earlier during a growth period are exponentially more abundant in the final population than clones from a mutation that occurs later. If the mutation rate was estimated by counting the resistant clones in just one culture, it would be heavily biased by *when* the resistance mutation(s) randomly occurred. Luria-Delbrück experiments account for this by counting resistant clones across multiple subpopulations. In QMS-seq, the subpopulations serve to mitigate the influence that early-occurring mutations could have on variability between replicates, and also to increase the likelihood of identifying rare or low-fitness mutations. Because the subpopulations are pooled prior to plating, any one mutation can only be found in a maximum of 1/32 of the total population (unless the same mutation occurred in multiple subpopulations). This means that during sequencing and variant calling (which has a detection threshold based on relative abundance), a more genetically diverse mutational landscape can be identified.

The three pooled populations were each spread across 10 selective MH agar plates supplemented with the MIC of a chosen antibiotic, using 200 µL of culture for each plate. We then incubated the 30 plates for three days to allow resistant mutants with severe fitness costs to form colonies. After three days, we collected and recombined colonies from the 10 plates into three heterogeneous resistant populations to be sequenced. There was an average of 130 colonies on each plate. Prior to collection, we manually removed sections of the largest colonies to equalize relative abundance and increase the likelihood of identifying rarer and low-fitness mutations. QMS-seq considers how many independent samples a mutation occurs in, rather than how prevalent it is within one sample population, so lowering the relative abundance of high-fitness mutants (which form large colonies) increases the relative abundance of low-fitness mutants, making them easier to detect without biasing later analysis.

To combine colonies from all 10 plates, we added fresh media to the plates and scraped them with a plate spreader to resuspend the cells. We mixed the media from all 10 plates and then thoroughly vortexed the tube to break up any clumps. We measured optical density at 600 nm to estimate cell count and diluted the populations to $4.0 \times 10^8$ cells/mL for storage at −80 °C with 15% glycerol. The samples were later thawed, and genomic DNA was extracted using the Genomic DNA Purification Kit (Promega). We determined the DNA concentration with a NanoDrop before sending it for sequencing.

## Whole-genome metagenomic sequencing and variant calling

We used Azenta Life Science's Illumina NovaSeq 2×150 base pair sequencing package. They performed sequencing to a mean depth of at least 1000x per sample, a value estimated to allow detection of mutations with a frequency of at least 0.5% within the population. For QMS-seq, raw sequencing reads were first trimmed with *Trim Galore* (v.0.6.7)[51] to remove sequencing adapters and low-quality reads, with a Q score cutoff of Q30. Trimmed reads were aligned to the reference genome (*E. coli* K-12 BW25113) with *Bowtie2 (v.2.2.5)*[52] using end-to-end alignment and the *−no-mixed* flag. BAM files were sorted with *samtools (v.1.16.1)*[53], and single-nucleotide variants (SNVs) with a minimum allele frequency of 1% were identified (these were considered "expected SNVs") using the *bcftools, mpileup, call*, and *view* commands. Prior to variant calling with *lofreq (v.2.1.3.1)*[6], BAM files were pre-processed based on GATK best practice protocol as recommended by the *lofreq* documentation. Specifically, the variant file was indexed with *GATK (v.4.2.6.1)*[54] *IndexFeatureFile* command, and BAM files underwent base quality score recalibration with the *BaseRecalibrator* and *ApplyBQSR* commands, using the expected SNVs from the previous step as known sites. Recalibrated BAM files were sorted with *samtools*, and low-frequency variants were called with *lofreq* with a significance threshold of 0.05 and using the *−call-indels* flag.

We took several precautions to reduce the possibility of miscalls. We ensured a high-quality alignment with the use of paired-end sequencing, stringent quality control of reads, and alignment of paired reads only while excluding unpaired reads. *lofreq* itself utilizes mapping quality scores and base scores (which were recalibrated to reduce systematic errors in the scores assigned during sequencing) in calling low-frequency variants. All samples had very high coverage and sequencing depth (Supplementary Table 2). We further checked the sequencing depth at the loci of the 3314 filtered mutation occurrences −the median depth for all variant sites was 1151x (Supplementary Fig. 4).

The pipeline described above can identify SNVs and small insertions or deletions of less than 30 base pairs, at very low frequencies in the population. To identify low-frequency mobilization events, which result in larger insertions and deletions, mutation calling was also performed using the *breseq* pipeline (v0.38.1)[7]. Trimming and read quality assessment was first performed on raw FASTQ files using *Cutadapt*[55] and *FastQC*[56] via the *Trim Galore* wrapper[51], according to default run parameters. Mapping and variant calling was performed using *breseq* polymorphism mode with a "polymorphism-frequency-cutoff" of 0.001, with other run parameters set to default values. Reads were aligned to a modified version of the reference genome for which we have additionally annotated mobile elements[49] as is required for the prediction of mobile element insertion events by *breseq*. Although *breseq* also calls SNPs, we did not use this output in our analysis because it was substantially less conservative, identifying many more mutations than *lofreq*.

## Controls to disregard media adaptations or variant calling errors

Our variant calling pipeline identified 2,191 unique mutations across the 35 samples, which corresponded to 6484 mutation occurrences since many mutations were identified across multiple samples. The mutations were incredibly diverse, targeting over 900 different genes, around 1/5 of the WT genome. We were concerned that our pipeline, designed to pick up very low-frequency variants in the population, had identified a host of additional mutations other than those providing antibiotic resistance. This seemed especially possible given the bacteria had three days of growth on the selective plate to accrue secondary mutations in addition to the resistance one.

Accordingly, we adopted multiple strategies to narrow our dataset down to only include mutations we could be highly confident were under positive selection for antibiotic resistance. First, we performed the same sequencing and variant calling pipeline as QMS-seq, with DNA from the WT strain that had grown in MH for only 12 h. Aligning reads from the WT genome against its own reference sequence identified any variant calls that resulted from misalignment[57] rather than a genuine mutation. Then, as a negative control, we performed three replicates of QMS-seq using the WT strain−but instead of plating on an antibiotic, we used non-selective agar. We also diluted the heterogeneous cultures prior to plating, to achieve a roughly equal number of colonies per plate, as we saw with the antibiotics. This checked for mutations that are under selection for better growth in rich media or on agar. From the original 2,191 mutations, 44 were also found in the 12-h samples, and another 126 were found in the negative control. These were excluded from subsequent analyses.

## Filtering mutations under clear selective pressure

To further increase confidence in the dataset of resistance mutations, we adopted a conservative filtering approach which identified mutations that were clearly under strong positive selection. Assuming a simplified possible mutation space of $1.4 \times 10^7$ (the number of bases in the WT genome times three possible mutations at each base), and given an average of 157 mutations occurring per sample, the likelihood of the exact same mutation occurring in 2 or more samples (out of 35) by random chance is:

$$1 - \left(1 - \frac{157}{1.4 \times 10^7}\right)^{35} - 35 \times \frac{157}{1.4 \times 10^7} \times \left(1 - \frac{157}{1.4 \times 10^7}\right)^{34} = 7.48 \times 10^{-8}$$

While the true possible mutation space is limited by essential bases and other factors, this number would not change by orders of magnitude. This calculation also excludes any other types of mutations, such as small insertions or deletions, and hence represents a conservative estimate. As such, we could be confident the 464 mutations that occurred in at least two samples were under strong positive selection.

This criterion was sufficient for intergenic and many genic mutations, however it excluded some genic mutations despite them being clustered in one gene. In certain genes, typically those targeted predominantly by nonsense or frameshift mutations, the *same* mutation would rarely appear in multiple samples, even if the gene was targeted by many *different* mutations across multiple samples. Hence, if there is insufficient pressure to mutate a specific base (or bases) in the gene over any other, the above criterion fails. To accommodate this, we also included 349 single-sample mutations observed in genes with more mutation occurrences than expected (Supplementary Fig. 5).

Our filtering criteria excluded 1208 of the remaining 2021 mutations, which reduced the number of targeted genes from 967 to 251. However, as our criteria generally filtered for mutations that occurred multiple times, the total number of occurrences only decreased from 4175 to 3314. This suggests that, while our filtering substantially decreased the observed variety of mutational targets, the filtered mutations still reflected most of the genetic diversity in the samples. To verify that the final 812 mutations we included in our analysis represented strong selective pressure, we compared their distribution throughout the genome to that observed by Foster et al. in a mutation accumulation study performed with *E. coli* in the absence of selection[58]. The gap sizes between mutations in our dataset indicated overwhelming clustering that was absent from the near-random distribution of 1625 mutations observed by Foster et al. (Supplementary Fig. 6).

Finally, we estimated the typical number of mutations present in a single genome by performing WGS on 18 individual clones that had been streaked from the glycerol stock of nine different heterogeneous resistant population samples. Raw sequencing reads were first

trimmed with *Trim Galore* with a Q score cutoff of Q20. Variants were then called with *snippy (v.4.6.0)*. This confirmed that most clones did not contain more than one mutation (Supplementary Table 1).

## Annotating regulatory features in the genome

We mapped intergenic mutations to regulatory features using information from published databases: regulonDB[59] and Ju et al.[60], containing positions of known promoters and Rho-dependent/independent terminators in *E. coli*. In some instances, regulatory annotations overlapped with a coding sequence. Mutations that occurred in such regions were classified as regulatory if they were near (<20 base pairs) the beginning or end of the gene. For the annotated promoters highlighted in Fig. 5A, we used BPROM[61] to identify their -10 and -35 sequences.

## Classifying known resistance and persistence genes

We searched Google Scholar with the following queries: (“*gene name*” + antibiotic resistance + mutation) and (“*gene name*” + antibiotic persistence) for the 251 target genes. For 49 annotated regulatory regions, each gene in the transcriptional unit was inserted for *gene name*. If a publication was found that indicated mutations in the gene were associated with antibiotic resistance, it was classed as such, regardless of the antibiotic or specific mutation(s). Antibiotic persistence genes were classified if they had specifically been implicated in persistence or found consistently dysregulated in persister populations. If no relevant publications appeared in the first 20 results, the gene was classed as not having been previously associated with antibiotic susceptibility.

## Categorizing the function of target genes

We used EcoCyc[62] to annotate every target gene with a known biological role, including those immediately downstream of regulatory targets. To classify the genes into broad functional categories, we used a combination of gene ontology (GO) biological process terms and published literature. For “membrane transport”, “translation”, “respiration”, “lipid metabolism”, “DNA synthesis/repair” and “transposase”, we identified the GO term(s) that were associated with each of these categories (Supplementary Data 4) and grouped target genes annotated with one of those terms. For transcription factors, which are annotated with GO terms indicating their enzymatic role rather than the downstream pathways they affect, classifications were supplemented with literature (Supplementary Data 2). Target genes were classified as “other” if they were annotated with GO biological process terms, but none fell under the above categories. They were classified as “uncharacterized” if they were not annotated with any GO biological process, and we could not find a description of their function in the literature. To construct an equivalent categorization of all the genes in the E. coli genome, we cross-referenced our list of GO terms against the GO annotations of the entire E. coli genome from PANTHER[63].

## Identifying matching mutations in the database of *E. coli* isolates

To contextualize QMS-seq mutations against global *E. coli* isolates, we downloaded all *E. coli* genomes from the BV-BRC database (as of April 2023)[18]. After filtering for high-quality data from Illumina platforms and excluding isolates that lacked sufficient metadata, we kept 21,711 genomes for downstream analysis, consisting of 4,918 raw sequence files and 16,793 assemblies. Variants were called using *snippy* with the same reference genome used above (NZ_CP009273.1), and we counted the number of isolates with sequences matching mutations in our dataset.

## Categorizing antibiotic-specific and strain-specific targets

Given the quantity of mutations and their complex distribution across 12 experimental conditions, we chose to be conservative when

categorizing mutational targets as “multi-drug resistance, MDR” versus “antibiotic-specific resistance, ASR”. Genes or regulatory features that were mutated in at least one sample from *all* antibiotics were classified as MDR. Those that were mutated in only one antibiotic were classified as ASR. For targets that were mutated in two antibiotics, we employed generalized linear modeling in *RStudio (v2023.09.1)* to statistically determine if they were disproportionately targeted in one ($P < 0.05$, after false discovery rate correction). Targets that were mutated in two antibiotics but without significant bias towards one, and any targets with less than three mutation occurrences, were classified as “unclear”.

Generalized linear modeling was also used to find targets whose mutational distribution varied significantly between strains. We identified mutational distributions with higher-order interactions between the antibiotic and strain variables and targets where antibiotic and strain were both significant but independent. We had the statistical power to identify targets with only strain effects (no antibiotic bias) but observed none. We used one-way ANOVA with multiple comparisons to examine differences in the total number of target genes (Fig. 4A) or mutations (Supplementary Fig. 3) between the 12 experimental conditions.

## Validating resistance genes

We used strains from the Keio knockout library to verify that mutations observed by QMS-seq provide resistance to the antibiotic they were selected by. We selected nine genes: four MDR and five ASR, which had been targeted by frameshift or stop-codon mutations across multiple samples. We then took the Keio strains with a deletion of the given strain and tested their ability to grow on MH agar containing 1x the MIC of CIP, CYC, or NIT. Four of the five ASR knockout strains were resistant to at least 1x the MIC of their respective antibiotic (Supplementary Fig. 7). The MDR knockout strains were all resistant to 1x the MIC of CIP and NIT, although two could not grow on CYC. While a meaningful comparison, a full gene knockout is not directly equivalent to a frameshift or stop-codon mutation, whose effects depend on how early in the polypeptide they occur.

## Mutation impact and dispersion analysis

Predicting the impact of genic mutations on the encoded protein was automated by SnpEff (v5.1)[64], which compares the variant sequences to the reference genome to predict the effect of single-nucleotide polymorphisms or small insertions/deletions. SnpEff categorizes mutations as high-impact (stop-codon insertions and frameshift indels), moderate-impact (non-synonymous substitutions and small in-frame indels), or low-impact (synonymous substitutions). To calculate the average mutation impact for every targeted gene, each impact type was given a numerical value: high = 1, moderate = 0.5, low = 0. An average was taken for all the mutation occurrences in one gene, and this value was used to put each gene somewhere on the three-color scale in Fig. 3B: red = 1, blue = 0.5, green = 0. We used Pearson’s Chi-squared test with Bonferroni correction and pairwise comparison to examine differences in the distribution of high/moderate/low-impact types in MDR versus ASR mutations.

We complemented this analysis by calculating the “mutation dispersion” for each gene, a metric quantifying how clustered or spread out the mutations were. Mutation dispersion was defined as the average distance (in base pairs) between adjacent mutations, divided by the maximum possible distance if all mutations were equally far apart. When calculating mutation dispersion, we discounted “outlier” mutations: low-occurrence mutations located distantly from a cluster of high-occurrence mutations. For each gene we considered every mutation occurrence as a number representing that mutation’s position between the first base of the gene and the last. We then calculated the interquartile range of these values and excluded mutations corresponding to outlier values. While still genuine resistance mutations,

excluding the outlier mutations from this analysis provides a dispersion value that better reflects the way the gene is most commonly targeted.

### Principal component analysis of experimental conditions

We set up a table with 12 columns for each experimental condition and 812 rows for all identified mutations, then gave every mutation a value between 0 and 3 for each condition corresponding to the number of replicates the mutation was identified in. We used the *factoextra (v1.0.7)* package to perform principal component analysis and visualize the variables distributed along the first two components.

### Genome visualization and functional site annotation

We used *Geneious Prime® (v2023.3.2)* to visualize mutations throughout the reference genome in relation to the positions of coding and regulatory sequences. The annotated genome displaying the location and occurrence value of every mutation is available in Supplementary Data 5. We used *InterPro*[65] to annotate experimentally-verified and computationally-predicted functional sites within target genes (such as *ykfM* and *gyrB*, Supplementary Figs. 1–2).

### Statistics and reproducibility

We used three samples per condition to allow estimating mutation re-occurrence across multiple samples while making testing many different antibiotic/strain combinations feasible. Randomization and blinding were used wherever possible. One sample from the NIT/Q513R condition was fully excluded from the analyses because the sequencing quality was low. All statistical tests are described in their relevant sections above.

### Reporting summary

Further information on research design is available in the Nature Portfolio Reporting Summary linked to this article.

## Data availability

Sequencing data has been deposited to NCBI with accession code PRJNA1071285 [https://www.ncbi.nlm.nih.gov/bioproject/1071285]. Detailed information about each of the mutations we analyzed is available in Supplementary Data 1. Source data for the figures are provided with this paper. Source data are provided with this paper.

## Code availability

Scripts used to analyze sequencing data are available on GitHub [https://github.com/Lagator-Group/QMS-seq/tree/main] under the terms of the GNU General Public License v3.

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

## Acknowledgements

The authors thank Patricia Barkoci, Rebecca Boccola, Michael Brockhurst, Elizabeth Fozo, Rowan Green, Deniz Ozbilek, and Torsten Seeman. The authors acknowledge assistance from Research IT and the use of the Computational Shared Facility at the University of Manchester. Funding was provided by a Wellcome Trust and the Royal Society Sir Henry Dale Fellowship (216779/Z/19/Z to M.L.), and the EPSRC Centre for Doctoral Training in BioDesign Engineering (EP/S022856/1 to M.J.J.).

## Author contributions

Conceptualization: M.J.J., J.K.S., S.D., M.L. Methodology: M.J.J., J.K.S., C.J.W., D.R.G., M.L. Investigation: M.J.J., J.K.S., D.R.G. Visualization: M.J.J. Funding acquisition: B.P.H., M.L. Supervision: B.P.H., M.L. Writing—original draft: M.J.J. Writing—review & editing: M.J.J., J.K.S., S.D., C.J.W., D.R.G., M.L.

## Competing interests

The authors declare no competing interests.
