## [Transparent Peer Review file · Nature Communications]

High-throughput method characterizes hundreds of novel antibiotic resistance mutations

Corresponding Author: Dr Mato Lagator

Version 0:

Reviewer comments:

Reviewer #1

(Remarks to the Author)

In this paper, Jago, Solely and colleagues develop a method for identifying mutations conferring antibiotic resistance (AR) to three unrelated antibiotics. The results show that many mutations were selected during the experiment, most of them not previously associated with antibiotic resistance. Overall, the paper is well-written, the figures are nice, and the results are well-presented and potentially interesting. However, I have a few concerns, mostly related to the author's interpretation of the results and the lack of benchmarking and validation of their method.

I am concerned that the paper does not provide a clear definition of what an antibiotic resistance mutation is. Although defining antibiotic resistance is complex, other papers have offered some definitions (see PMID: 25534811) that the authors might find useful. For example, mutations in DNA repair proteins are likely to affect intrinsic resistance levels (more on this below), but caution should be exercised in claiming that these mutations confer antibiotic resistance. This caution is especially warranted given that i) the authors did not test the individual contribution of the mutations to AR, ii) selection was performed at low antibiotic concentrations (1xMIC), and iii) the protocol cannot distinguish between potential AR mutations and compensatory mutations that emerged during plating. It seems somewhat far-fetched to claim that the 975 mutations (found in 278 genes) are antibiotic-resistance mutations, and a more nuanced interpretation is definitely needed. Besides, the paper's message could be strengthened by validating some of the mutations found (the case studies?). This would entail producing clean mutants and measuring the MIC. Some mutations could be tested easily using the KEIO collection, so it should be relatively straightforward.

My second concern relates to the method presented by the authors (QMS-Seq). The main novelty of the method is splitting a founder population into 32 subpopulations and allowing each subpopulation to accumulate mutations. According to the authors (L457), this step increases diversity and the relative frequency of low-fitness mutations by limiting founder effect bias or selective sweeps. Was this quantified? How much was diversity increased by performing this step? This is important because the paper (e.g., title, abstract, discussion) focuses on the method itself, and its performance is not evaluated, nor is it compared with classic methods (plating a single culture), raising doubts about the novelty and suitability of the method.

Specific comments:

What was the rationale for choosing the antibiotics?

The authors first state that they found hundreds of AR mutations, but then they show that a sizeable fraction are likely persistence-related mutations. Resistance and persistence (and tolerance) are different phenomena and should not be used interchangeably (see, for instance, PMID: 27080241).

Figure 2A. Why are some mutations present in (almost) all 35 populations? For instance, *lpxC*, *gatY*, *stfP*... Are these considered antibiotic-resistance (MDR) mutations? What's the author's explanation for that many mutations across populations?

Figure 2b. The line connecting Intergenic with their regulatory features is hard to spot because it is "behind" the non-synonymous box. Could this be improved by drawing the line in the free space on the right-hand side of the figure?

L136. This statement needs statistical support.

Are *recO* and *uvrB* considered resistance mutations? A strict definition of resistance mutations will greatly benefit the paper. Many mutations listed in the Supp. Table affects metabolic genes, which, as the authors state, have never been related to antibiotic resistance.

L211: I agree that the effect of the *rpoB* genotypes somewhat affects the number and identity of the resistance mutations. Yet, the PCA analyses show that most genotypes cluster by antibiotic (except CIPQ513R). This indicates that most variation is not attributed to genetic background. Therefore, I recommend the authors to tone down sentences like L211-212. L230-L244. This paragraph is a tad unclear. Please add more context or rewrite it so an average reader unfamiliar with *aceF* (like myself) can understand it. I particularly struggled with L234-238. According to the methods section, the authors employed two complementary methods to call mutations: *breseq* and *lofreq*. I understand that *breseq* was employed for larger mobilisation events, but the typical *breseq* output includes SNPs and other mutations. I wonder whether *lofreq* and *breseq* results were concordant. How many mutations were called by both approaches? I was surprised by the abundance of some of the mutations used as case studies. For instance, the *aceF* and *sis* loci are targeted more than 200 times each. I wonder if this overrepresentation could be due to calling/assembly errors, particularly given that some of these sequences are repeated within the chromosome. A good control that is lacking is running the variant calling algorithm (*lofreq/breseq*) using sequencing reads against a reference genome assembled with those same reads. In this case, it would mean using BW25113 reads against the assembled BW25113 reference. By mapping BW25113's sequencing reads against itself, the errors made by the assembly/variant calling algorithm can be identified and subtracted from subsequent analysis.

(Remarks on code availability)

Reviewer #2

(Remarks to the Author)

In the manuscript « High-throughput method rapidly characterizes hundreds of novel antibiotic resistance mutations », a method combining high diversity generation, screen on selective media and deep-sequencing is presented, which allows for the identification of the mutational target for resistance to three different antibiotics. The manuscript is notably well prepared, well written and the figures are creative, illustrate perfectly the results and perfectly convey the main messages of the paper. I have few concerns that the authors should address before I can recommend this manuscript for publication :

- (1) The claim that the mutations identified arose in independent populations is a bit of an overstatement, as, if I understand well, for each genetic background, there was only one preculture from the glycerol stock. Although it is difficult to evaluate the number of division occurred in this preculture, it is reasonable to think that it is above 5, whereas the number of division during the independent phase (after the preculture has been split into 96 sub-populations) is around 13. So, more than a quarter of the divisions leading to the final diversity are common to all populations for a same background. This non-independence through the preculture is common to many experimental evolution approach but here it is a bit more of a problem (see my next point) because it represents a relatively large part of the evolutionary history explored.
- (2) The experimental design is missing a negative control which would have consisted in plating the 12 pools on non-selective plates at a dilution which allows to collect roughly the same number of colonies as on the 10 selective plates in the other treatment, followed by sequencing. This would allow
 - a. to detect mutations beneficial in the specific conditions of the experiment (Mueller Hinton medium, fast growth...) or mutations that arose during the preculture step and so were present across many of the 96 individual populations for each condition. This is important because these mutations are interpreted as MDR mutations with the data at hand.
 - b. To identify potential mapping errors in homologous regions (see point 4 below).
- (3) It would be very helpful to have an estimate of the number of colonies which were pooled in each sample before sequencing. This would inform on the likelihood of a mutation to be detected depending on the number of colonies in which it is present (knowing the sequencing depth and the detection threshold) and to verify that the total number of mutations obtained is coherent with the « distribution » of number of mutations per colony presented in extended data table 1.
- (4) Some of the mutations detected could be artefacts due to mis-mapping among homologous regions. This concern was raised by the example the authors develop on *aceF* and in particular by their sentence « surprisingly all mutations occurred at positions that lack consensus between the three sites, invariably serving to bring one site into better alignment with another ». It is actually frequent to get mis-mapping of one homologous region on another with *bowtie2* (this is quite often the case with the 7 ribosomal RNA operons in *E. coli* which are very homologous and reads from one operon is mapped on another operon and artefact mutations are detected). The fact that the mutations detected here always correspond to one of the other versions of the region points to this kind of error. The authors should be able to check whether these mutations are real or artefacts by looking at the linkage between them in the individual reads mapped on this region as the « mutated » position are sufficiently close in the sequence to be on the same read. This concern also holds for *sib-ibs* and potentially for other mutations identified in homologous regions. If these mutations turned out to be artefacts, this would require cleaning the data set from them and redoing a large part of the analyses but it would not remove the global interest and soundness of the study.
- (5) The mutation dispersion metrics used does not correct for the number of unique mutations in the gene. It would be better to use the ratio of observed / expected by chance of average distance between mutations, knowing the length of the gene and the number of unique mutations in the gene. This would also avoid the bidimensional representation in figure 3A and 3B, which is not particularly easy to interpret.
- (6) Fig 3D: Have the authors conducted proper enrichment analysis?
- (7) L628-631 : was correction for multiple testing applied?

Minor comments :

(1) I know the space constraint is quite strong in this paper format, but I find the end of the introduction and the beginning of the results are missing key informations (which are present in the material and methods). For example, the authors should give more info on the four genetic backgrounds and the mechanism of action of the three antibiotics (I62-63) or refer to extended data table 1 when they mention « variants with only a single mutation » (I52).

(2) L48 : I see the protocol developed more as an extension of a Luria-Delbruck / mutant screen protocol (such as the one applied in Harmand et al. 2017, for example) than an experimental evolution approach as « evolution » only occurs for 48h and that during these 48h mutation is the main evolutionary force (as the environment is mild and selection weak). Could the authors rephrase this section to be more accurate ?

(3) Figure 2B : it is quite unfortunate to use the term « silent » for synonymous mutations given the results exposed afterwards (and results from many other groups showing that synonymous mutations are not always neutral/silent).

(Remarks on code availability)

Version 1:

Reviewer comments:

Reviewer #1

(Remarks to the Author)

The authors have made significant improvements to the manuscript, and I am satisfied with most of the changes. In particular, I believe that the new analysis of the mutations reduces most of the noise and strengthens the overall results.

Regarding the new case studies, I agree that they show the variety of mechanisms uncovered by QMS-seq; however, they do come across as somewhat speculative. For example, the authors claim that lpxC mutations are associated with colistin resistance (L275). However, it's important to note that lpxC mutations only confer colistin resistance in MCR-carrying bacteria. In wild-type bacteria, these mutations lower the colistin MIC (i.e., they're epistatic; see ref.36). Additionally, it remains unclear how the lpxC mutations that confer resistance to colistin would provide resistance to CIP, NIT, or CYC.

Therefore, I recommend revising the section title, "Case studies demonstrate the depth of mechanistic insights afforded by QMS-seq," to something more reflective of this section's speculative nature, perhaps along the lines of L71.

(Remarks on code availability)

Reviewer #2

(Remarks to the Author)

The authors have performed extra experiments and revised many sections of the paper to account for the reviewer's comments and suggestions. The clarity of the manuscript has been improved and the authors have unveiled new pieces of information through the additional experiments.

All my concerns have been adequately addressed.

I have just a very minor comment regarding the new "case studies" they present: for two out of the three examples of synonymous mutations, the mutations are actually located at the very beginning of the genes and in this portion of the gene, it has been shown that the effect of synonymous mutations is often due to the effect on mRNA secondary structure and access of the ribosome to the Ribosomal Binding Site. In other words, these mutations might affect the level of translation of these genes. This explanation could be added to the manuscript around I290.

(Remarks on code availability)

Reviewer #1

In this paper, Jago, Solely and colleagues develop a method for identifying mutations conferring antibiotic resistance (AR) to three unrelated antibiotics. The results show that many mutations were selected during the experiment, most of them not previously associated with antibiotic resistance. Overall, the paper is well-written, the figures are nice, and the results are well-presented and potentially interesting. However, I have a few concerns, mostly related to the author's interpretation of the results and the lack of benchmarking and validation of their method.

1. I am concerned that the paper does not provide a clear definition of what an antibiotic resistance mutation is. Although defining antibiotic resistance is complex, other papers have offered some definitions (see PMID: 25534811) that the authors might find useful. For example, mutations in DNA repair proteins are likely to affect intrinsic resistance levels (more on this below), but caution should be exercised in claiming that these mutations confer antibiotic resistance. This caution is especially warranted given that i) the authors did not test the individual contribution of the mutations to AR, ii) selection was performed at low antibiotic concentrations (1xMIC), and iii) the protocol cannot distinguish between potential AR mutations and compensatory mutations that emerged during plating. It seems somewhat far-fetched to claim that the 975 mutations (found in 278 genes) are antibiotic-resistance mutations, and a more nuanced interpretation is definitely needed.

We agree with the Reviewer that our definition of resistance needed more clarity and caution. We made changes to the text (L186-91) to clarify the definition of resistance mutation. Namely, we consider resistance mutations ones that have passed our criteria for strong positive selection due to antibiotic exposure. This meant that they emerged in strains that were capable of growth at a concentration of an antibiotic that the wildtype could not tolerate. In other words, all mutations we call 'resistance mutations' occurred in strains that exhibited a measurable change in MIC. We validated that the majority of mutants in our experiment had 1-2 mutations, meaning that at least the majority of all resistance mutations we called in our study conferred that measurable change in MIC. This is especially true given that 60% of the mutations originally identified by the variant calling pipeline were excluded by our filtering.

Our filtering criteria (L539) ensured that our final dataset of mutations only contained those that either: (i) occurred in at least two independent experimental samples or (ii) were found within genes that were targeted by significantly more mutations than expected across multiple samples. This provided further confidence that the mutations in our dataset did not occur by chance and rather were a consequence of strong selection.

However, it is possible that some of the identified mutations could have occurred in an already mutated background that carried a resistance mutation. In particular, this could have occurred during the last step of colony growth on selective plates, where a strain carrying resistance mutation could

have acquired additional mutations that, for example, compensate for the cost of resistance. This might be particularly relevant in one of the strain/antibiotic combinations, which exhibited elevated mutation rates combined with carrying mutations in the DNA repair genes. These mutator strains could have acquired a mutation in DNA repair and only subsequently acquired the resistance mutation. However, because these mutations occurred in at least two independent samples which is extremely unlikely to occur by chance ($\sim 10^{-7}$) without strong selective advantage, we think that capturing these mutations is still valuable, as they reflect an accurate picture of the tools available to bacteria when evolving resistance.

We made changes to the text to be more precise about the definition of resistance we used and its consequences on understanding our dataset (L370-385). This included clarifying that the aim of QMS-seq is not to identify a list of mutations that *all* provide 'true' antibiotic resistance and alter MIC. But rather to comprehensively identify a landscape of mutations (and genes) that are under positive selection during evolution of resistance to a given antibiotic. We also mention this in the Introduction (L56)

Regarding the Reviewer's specific concerns: i) we validated the level of resistance of various mutants showing they do have a higher MIC (elaborated in comment 2); ii) we expanded the discussion to address the influence our chosen concentration (1x MIC) had on the types of mutations we observed; iii) we elaborate further on the possibility of having detected compensatory and other additional mutations.

2. Besides, the paper's message could be strengthened by validating some of the mutations found (the case studies?). This would entail producing clean mutants and measuring the MIC. Some mutations could be tested easily using the KEIO collection, so it should be relatively straightforward.

We appreciated the suggestion to utilize the Keio collection, which made for excellent proxies to evaluate the antibiotic resistance provided by QMS-Seq mutations. We chose five strains for antibiotic-specific genes, corresponding to every gene in the top right quadrant of Fig. 3A. They are overwhelmingly targeted by high impact mutations and appear across multiple samples. Four of the five were resistant to 1X the MIC of their respective antibiotic.

We also chose four strains for the multi-drug genes, which had been repeatedly targeted by specific frameshift or stop-codon mutations (in contrast with the majority of MDR genes, which were targeted by non-synonymous mutations). All these strains were resistant to 1X the MIC of CIP and NIT, however two were not resistant to 1X MIC of CYC.

There are several possible reasons why certain Keio knockout strains were not resistant, despite their gene being targeted by knockout mutations in the QMS-seq dataset. Primarily, while it offers an easy comparison, a gene being fully excised from the genome and replaced with a kanamycin resistance cassette (as in the Keio strains) is not the same as a gene being knocked out by a frameshift or stop codon mutation. Depending on where such a mutation

occurs, large parts of the protein (upstream of the mutation) can still be produced correctly. In the case of frameshift mutations, regardless of the mutation's location a novel polypeptide tail can be produced, with unpredictable phenotypic effects. Furthermore, inserting a large cassette instead of just a frameshift mutation can have consequences on bacterial fitness. The observed lack of CYC resistance in *ubiG* was particularly surprising, as many *ubi* genes including *ubiG* were implicated in resistance to cycloserine in a previous study, which screened the Keio collection for antibiotic resistant strains (PMID: 23316042).

We summarized these findings in the Results (L183-186) and provided more detail in the Methods (L646-656).

Figure legend: (A) Figure 3A from the main text, annotated to show which genes we picked to validate their role in resistance using the Keio collection. (B) Whether the Keio strain lacking the given gene was resistant to 1X the MIC of the antibiotics that, during QMS-seq, selected for knockout mutations in the same gene.

- My second concern relates to the method presented by the authors (QMS-Seq). The main novelty of the method is splitting a founder population into 32 subpopulations and allowing each subpopulation to accumulate mutations. According to the authors (L457), this step increases diversity and the relative frequency of low-fitness mutations by limiting founder effect bias or selective sweeps. Was this quantified? How much was diversity increased by performing this step? This is important because the paper (e.g., title, abstract, discussion) focuses on the method itself, and its performance is not evaluated, nor is it compared with classic methods (plating a single culture), raising doubts about the novelty and suitability of the method.

We thank the Reviewer for this comment, as it made clear we had poorly explained some aspects of QMS-seq's experimental design, particularly this one. Splitting into 32 sub-populations during the mutation accumulation step is not intended to increase genetic diversity in the traditional sense (*i.e.*, increasing the number of distinct alleles/mutants in the population). It instead helps equalize the relative abundance of distinct alleles/mutants – making it easier to *identify* a more genetically diverse collection of mutants after sequencing the sample with a set detection threshold. It should also substantially reduce variability between replicates, as the mutational landscape is less affected by the random variable of when mutations occur. The practice is well established in Luria-Delbruck experiments / fluctuation assays, which use parallel cultures to reduce variability when measuring mutation rates (PMID: 31840662). We added a paragraph to the Methods which describes the logic behind using subpopulations in full (L442-454), along with changes to the main text (L338-340).

We believe the main novelty of the method comes from the application of metagenomic sequencing to populations evolved over a short-term, combined with the high-throughput design that enabled us to use re-occurrence across samples as a quantitative metric of likelihood. We made changes to the manuscript to better emphasize this novelty (L343-348, 367-368).

4. What was the rationale for choosing the antibiotics?

We chose the three antibiotics because they have different mechanisms of action; they are used in treating *E. coli* infection (although cycloserine irregularly); resistance to them is often inherited through point mutations rather than the horizontal transfer of resistance genes; and many resistance mutations to them in *E. coli* have been characterized. We have updated the introduction to include our rationale behind antibiotic selection, as well as briefly stating their mechanisms of action (L66-69). We have added a more thorough explanation to the methods (L413-422).

5. The authors first state that they found hundreds of AR mutations, but then they show that a sizeable fraction are likely persistence-related mutations. Resistance and persistence (and tolerance) are different phenomena and should not be used interchangeably (see, for instance, PMID: 27080241).

As described in comment 1, we have made multiple changes aiming to be more precise about the definitions of 'resistance' and 'resistance mutation'. Under our definition, heritable mutations in genes typically associated with persistence would be considered 'resistance mutations'. We consider this to be an important finding in our study – that heritable mutations in persistence genes (including in their regulatory sequences) could provide a similar phenotype to them being dysregulated during persistence: very slow growth and antibiotic tolerance. Indeed, there were many colonies with slow growth rates on the antibiotic plates, which was chiefly why we incubated the plates for 72 hours. To better capture this argument, we have also made changes to the Discussion (L360-365) clarifying our hypotheses about why we observed

mutations in genes that have previously been associated with antibiotic persistence.

6. Figure 2A. Why are some mutations present in (almost) all 35 populations? For instance, *lpxC*, *gatY*, *stfP*... Are these considered antibiotic-resistance (MDR) mutations? What's the author's explanation for that many mutations across populations?

We consider these MDR mutations, that enable growth on 1X the MIC of multiple antibiotics. There were roughly 1300 colonies in the combined populations sequenced for each sample, meaning it's likely many colonies from the ~45,500 colonies collected across all 35 samples would have had the same resistance mutation. The response to comment 16 has a more detailed explanation of this.

We have confidence in these mutations conferring resistance, as 8 of the 12 genes that were targeted in >30 samples have previously been associated with antibiotic resistance. Mutations in *lpxC* specifically are known to provide resistance to colistin (PMID: 35943060), which is now mentioned in the text as part of one of the new case studies. One of the genes we validated for resistance using the Keio collection (see comment 2) was *ydgH*, which was targeted in 32 samples and has previously been implicated in multi-drug resistance (34491801). We confirmed that a strain lacking this gene was resistant to at least 1X MIC of CIP, CYC, and NIT.

Therefore, we are confident that the genes which appeared in the vast majority of QMS-seq samples are genes that are commonly targeted to provide resistance to 1X the MIC of different antibiotics.

7. Figure 2b. The line connecting Intergenic with their regulatory features is hard to spot because it is "behind" the non-synonymous box. Could this be improved by drawing the line in the free space on the right-hand side of the figure?

We thank the Reviewer for this suggestion and have made the change.

8. L136. This statement needs statistical support.

We performed a two-sample Wilcoxon ranked sum test on mutation dispersion values to show MDR genes are significantly different than ASR genes, and a one-sample Wilcoxon signed-rank test to show MDR values are overwhelmingly lower than 15%. We changed the text to reflect these changes (L155-157).

9. Are *recO* and *uvrB* considered resistance mutations? A strict definition of resistance mutations will greatly benefit the paper. Many mutations listed in the Supp. Table affects metabolic genes, which, as the authors state, have never been related to antibiotic resistance.

See response to comment 1, describing in detail our definition of 'resistance mutation'. The CIP/Q513-specific *recO* and *uvrB* mutations are now

mentioned in the Discussion to suggest their likely role in facilitating acquisition of genuine resistance mutations.

Regarding the many mutations in metabolic genes: mutations targeting metabolism genes (PMID: 33602825) and their regulatory regions (39180403) are frequently involved in clinical resistance in *E. coli*. While many genes we identified have not been related to resistance previously, our study was designed to explore less well characterized resistance mutations. In the Discussion, we made much clearer the reasons we think led to us identifying so many genes not previously associated with resistance (L337-351).

10. L211: I agree that the effect of the *rpoB* genotypes somewhat affects the number and identity of the resistance mutations. Yet, the PCA analyses show that most genotypes cluster by antibiotic (except CIPQ513R). This indicates that most variation is not attributed to genetic background. Therefore, I recommend the authors to tone down sentences like L211-212.

We have removed the more emphatic language from these sentences (L238-240).

11. L230-L244. This paragraph is a tad unclear. Please add more context or rewrite it so an average reader unfamiliar with *aceF* (like myself) can understand it. I particularly struggled with L234-238.

After performing additional controls we have removed the *aceF* case study (see comment 13). We removed these to be conservative in the set of mutations we called as being 'resistance mutations' – for more detailed explanation, please see response to comment 13. We replaced the *aceF* and *sib-ibs* case studies with other examples of interesting mutational targets.

12. According to the methods section, the authors employed two complementary methods to call mutations: *breseq* and *lofreq*. I understand that *breseq* was employed for larger mobilisation events, but the typical *breseq* output includes SNPs and other mutations. I wonder whether *lofreq* and *breseq* results were concordant. How many mutations were called by both approaches?

The *breseq* results were substantially less conservative than the *lofreq*, as it identified nearly 10x more SNPs and small indels compared to *lofreq* (21,330 vs 2,211). *Lofreq* is specifically designed to call mutations with a very low error rate from metagenomic sequencing data with high depth, while *breseq* is much less specialized. As such, while most of the mutations identified by *lofreq* were also identified by *breseq*, we chose to exclusively use the *lofreq* output for SNPs and small indels. We added a sentence to the methods to clarify this (L512)

13. I was surprised by the abundance of some of the mutations used as case studies. For instance, the *aceF* and *sis* loci are targeted more than 200 times each. I wonder if this overrepresentation could be due to calling/assembly errors, particularly given that some of these sequences are repeated within the chromosome. A good control that is lacking is running the variant calling algorithm

(lofreq/breseq) using sequencing reads against a reference genome assembled with those same reads. In this case, it would mean using BW25113 reads against the assembled BW25113 reference. By mapping BW25113's sequencing reads against itself, the errors made by the assembly/variant calling algorithm can be identified and subtracted from subsequent analysis.

We thank the Reviewer for this suggestion, which proved an effective way to ensure our dataset is conservative and to answer the question of mutations in the homologous regions. We performed the following experiment: growing the WT strain for 12 hours, before isolating gDNA and sending for sequencing with the same parameters as used in QMS-Seq. We ran this data through the QMS-Seq variant calling pipeline, and identified 168 mutations, 44 of which were also present in our original dataset of 2,191 mutations (not filtered for mutations under strong selective pressure). While many of the 168 mutations (i.e., errors) did occur in tandem repeat regions with high percentage of homology to nearby sequences, they were not nearly as frequent within these regions as the mutations we identified in the original dataset. These results made it clear that our variant calling pipeline is capable of making mistakes when aligning homologous regions, but indicated the pattern of mutations we had originally observed was not solely due to alignment errors.

To shed further light on this problem, we performed a full negative control of the QMS-Seq experiment. This consisted of taking the WT strain, and doing the first steps of QMS-seq the same way: split into 3 replicate population, overnight growth, dilute 1:10,000, grow for 24 hours in 3 x 32 subpopulations, recombine into 3 heterogenous populations. Then, rather than plating on 3 x 10 plates containing agar with an antibiotic, we plated on 3 x 10 non-selective agar plates. We also diluted the culture prior to plating to achieve a roughly equal number of colonies per plate as had been observed on the antibiotic agar (~130). We then collected, sequenced, and analysed these samples as we had with the original experiment. The variant calling pipeline identified 541 mutations, 126 of which were also present in the original dataset of 2,191 mutations.

This time, the pattern of mutations in homologous regions had much greater overlap with what we had observed originally with the antibiotic plates. Every single "mutation" occurred at a position within a repeat that was not the same base as the equivalent position in the other repeat(s). We looked at the linkage of these mutations, by examining the individual sequencing reads that they were called from. This showed that, by and large, the SNP mutations at variant positions all occurred together on individual reads. This strongly suggests the mutations were called because of reads being misaligned to a different repeat sequence.

However, the fact that these mutations were substantially more common in the negative control compared to sequencing the WT suggests these results do not solely come from misaligning WT reads to their own reference genome. We believe the most likely explanation is that the repeat regions with many "mutations" are areas of the genome with a high frequency of duplication during evolution under minimal selection. Such duplications, which

could easily vary slightly from their original sequence, would be even harder to align properly during variant calling, as no perfect match would be present in the reference genome. If such duplications occurred with high frequency during the 24h mutation accumulation, they would be present in some of the colonies across all samples, explaining the pattern of miscalls that we observed.

This still represents an interesting result, as the majority of repeat regions within the genome did not appear mutated in any of these experiments, indicating there are substantial differences in how likely different repeat regions are to duplicate further. However, further exploring this is outside the scope of this work. Consequently, we have removed ALL mutations from the dataset that occurred within repeat regions (not only mutations that were identified in the 12h-WT and negative control experiments). This reduced our dataset of mutations filtered for strong positive selection from 975 mutations to 813 mutations. We reanalyzed the dataset and recreated every figure, but none of our major findings changed. However, this did necessitate removal of the *aceF* and *sib/ibs* case studies.

We have replaced these case studies with interesting examples of genes targeted by different types of regulatory mutations (L267-300). We describe these experiments in the Methods (L526-537).

Figure legend: Mutations in *aceF* repeat regions identified by: original QMS-seq experiment, coloured by number of samples / mutation occurrence (top), sequencing a WT culture grown for only 12 hours (middle), negative control of QMS-seq, plating on non-selective agar (bottom).

Reviewer #2

- The claim that the mutations identified arose in independent populations is a bit of an overstatement, as, if I understand well, for each genetic background, there was only one preculture from the glycerol stock. Although it is difficult to evaluate the number of divisions occurred in this preculture, it is reasonable to think that it is above 5, whereas the number of divisions during the independent phase (after the preculture has been split into 96 sub-populations) is around 13. So, more than a quarter of the divisions leading to the final diversity are common to all populations for a same background. This non-independence through the preculture is common to many experimental evolution approaches but here it is a bit more of a problem (see my next point) because it represents a relatively large part of the evolutionary history explored.

We apologise for the lack of clarity in our original submission regarding this point. Dividing the inoculate into three independent populations was done prior to initial overnight growth (i.e. preculture). As such, no cell divisions should be common to all three replicate populations for each condition. We updated the Methods to be more explicit (L433-436).

15. The experimental design is missing a negative control which would have consisted in plating the 12 pools on non-selective plates at a dilution which allows to collect roughly the same number of colonies as on the 10 selective plates in the other treatment, followed by sequencing. This would allow
 - a. to detect mutations beneficial in the specific conditions of the experiment (Mueller Hinton medium, fast growth...) or mutations that arose during the preculture step and so were present across many of the 96 individual populations for each condition. This is important because these mutations are interpreted as MDR mutations with the data at hand.
 - b. To identify potential mapping errors in homologous regions (see point 4 below).

We thank the Reviewer for pointing out this oversight, and their suggestion which provided very informative results. We completed the negative control as described: performing QMS-seq with the WT strain, but with a dilution prior to plating and using agar with no antibiotic.

This helped identify potential media adaptations and alignment errors, so we could remove them from the dataset. Please see comment 13 for a full account of the results from this and another control experiment, and how the manuscript has changed after further filtering.

We've added a full description of the negative control experiment to the methods, and its utility in removing errors and mutations that are not related to antibiotic selection (L516-537).

16. It would be very helpful to have an estimate of the number of colonies which were pooled in each sample before sequencing. This would inform on the likelihood of a mutation to be detected depending on the number of colonies in which it is present (knowing the sequencing depth and the detection threshold) and to verify that the total number of mutations obtained is coherent with the « distribution » of number of mutations per colony presented in extended data table 1.

While the number varied between antibiotics and individual plates, there was roughly an average of 130 colonies on each plate, meaning 1300 colonies in each sample. We have included this information in the methods (L460). Prior to filtering, there was an average of 157 mutations detected in each sample, with the lowest frequency variants detected at 0.3% of the population. If we assume equal number of cells in each colony, a mutation would need to be present in ≥ 4 colonies (0.3% of 1300) in order to be detected. While we attempted to mitigate this by removing sections of the largest colonies, there were substantial differences in the sizes of colonies and thus the amount of DNA they contribute to the final sample – influencing these calculations. Many of the smaller colonies may have contributed less than 0.3% of the final

sample, hence going undetected. This would help explain the difference in numbers of colonies sequenced and numbers of mutations identified.

The distribution from extended data table 1 (based on 18 colonies we selected for whole genome sequencing) has a median of one, and maximum of two. This suggests multiple mutations from one individual colony being detected is unlikely, which also makes sense given we identified far fewer mutations than colonies.

17. Some of the mutations detected could be artefacts due to mis-mapping among homologous regions. This concern was raised by the example the authors develop on *aceF* and in particular by their sentence « surprisingly all mutations occurred at positions that lack consensus between the three sites, invariably serving to bring one site into better alignment with another ». It is actually frequent to get mis-mapping of one homologous region on another with bowtie2 (this is quite often the case with the 7 ribosomal RNA operons in *E. coli* which are very homologous and reads from one operon is mapped on another operon and artefact mutations are detected). The fact that the mutations detected here always correspond to one of the other versions of the region points to this kind of error. The authors should be able to check whether these mutations are real or artefacts by looking at the linkage between them in the individual reads mapped on this region as the « mutated » position are sufficiently close in the sequence to be on the same read. This concern also holds for sib-ibs and potentially for other mutations identified in homologous regions. If these mutations turned out to be artefacts, this would require cleaning the data set from them and redoing a large part of the analyses but it would not remove the global interest and soundness of the study.

We used two experiments to determine if mutations were called due to mis-mapping: (i) sequencing wildtype reads and aligning them to their own reference genome (ii) the negative control, plating on non-selective agar. While we believe that many of the mutations in homologous regions may be a consequence of rapid amplification events, we could not be certain of that so we removed all of them from the dataset. Please see comment 13 for a full explanation of the experiments we performed to confirm this, including checking the linkage. We greatly appreciate the Reviewer's comment regarding the study's interest and soundness.

18. The mutation dispersion metrics used does not correct for the number of unique mutations in the gene. It would be better to use the ratio of observed / expected by chance of average distance between mutations, knowing the length of the gene and the number of unique mutations in the gene. This would also avoid the bidimensional representation in figure 3A and 3B, which is not particularly easy to interpret.

We agreed that the suggested calculation is stronger, and after re-making the plots were shocked to find them identical to the previous. Unintuitively, it seems that the two calculations are actually equivalent.

We changed the text to use the new equation (L191, 671-674), as we agree with the Reviewer that it is more intuitive. We did however keep the structure of Fig.3A/B, as we think they convey the core message of the paper succinctly.

Here, we include the mathematical proof that the two definitions are equivalent:

Assuming a gene of length L with n mutations, and every mutation's position in the gene is given by $\{m_1, m_2, \dots, m_n\}$

Original equation:

$$\frac{\text{bp of last mut} - \text{bp of first mut}}{\text{gene length (bp)}} = \frac{m_n - m_1}{L}$$

New equation:

$$\frac{\text{avg distance between adjacent muts}}{\text{max possible avg distance}} = \frac{\sum_{i=2}^n (m_i - m_{i-1}) \div (n - 1)}{L \div (n - 1)}$$

The average gap distance between adjacent mutations is given by the sum of the distances divided by the number of gaps ($n - 1$). The maximum possible average gap distance (i.e. 100 on the dispersion axis) occurs when one mutation is at the first base, one is at the last base, and any additional mutations are spread evenly. This equates to the gene length divided by the number of gaps ($n - 1$). The two ($n - 1$) cancel out, and because the sum of gap distances is equal to the last mutation's bp minus the first mutation's bp, the original equation is replicated.

19. Fig 3D: Have the authors conducted proper enrichment analysis?

We added significance stars from gene set enrichment analysis to the figure (L144).

20. L628-631: was correction for multiple testing applied?

We performed multiple testing correction using false discovery rate (FDR) correction but had mistakenly not made this clear in the methods. We thank the Reviewer for pointing this out and have rectified the mistake (L633).

21. I know the space constraint is quite strong in this paper format, but I find the end of the introduction and the beginning of the results are missing key informations (which are present in the material and methods). For example, the authors should give more info on the four genetic backgrounds and the mechanism of action of the three antibiotics (l62-63) or refer to extended data table 1 when they mention « variants with only a single mutation » (l52).

At the end of the introduction, we added a paragraph describing the four genetic backgrounds and the antibiotics' mechanism of action, as well as the

study's design and purpose more broadly (L60-72). We also added a more detailed description of the antibiotics' mechanisms and our rationale behind choosing them to the methods (L413-422).

22. L48 : I see the protocol developed more as an extension of a Luria-Delbruck / mutant screen protocol (such as the one applied in Harmand et al. 2017, for example) than an experimental evolution approach as « evolution » only occurs for 48h and that during these 48h mutation is the main evolutionary force (as the environment is mild and selection weak). Could the authors rephrase this section to be more accurate ?

This is a good point, and the Reviewer discerned our original inspiration for the QMS-seq protocol (Luria-Delbruck assays), which we had neglected to mention in the manuscript. We have rephrased the Introduction to replace “experimental evolution technique” with “mutant screening technique” (L45). We also added a paragraph to the Methods explaining how Luria-Delbruck assays inspired specific aspects of our experimental design (partly in response to comment 3) (L442-454).

23. Figure 2B : it is quite unfortunate to use the term « silent » for synonymous mutations given the results exposed afterwards (and results from many other groups showing that synonymous mutations are not always neutral/silent).

We replaced “silent” with “synonymous”.

Reviewer #1

The authors have made significant improvements to the manuscript, and I am satisfied with most of the changes. In particular, I believe that the new analysis of the mutations reduces most of the noise and strengthens the overall results.

Regarding the new case studies, I agree that they show the variety of mechanisms uncovered by QMS-seq; however, they do come across as somewhat speculative. For example, the authors claim that *lpxC* mutations are associated with colistin resistance (L275). However, it's important to note that *lpxC* mutations only confer colistin resistance in MCR-carrying bacteria. In wild-type bacteria, these mutations lower the colistin MIC (i.e., they're epistatic; see ref.36). Additionally, it remains unclear how the *lpxC* mutations that confer resistance to colistin would provide resistance to CIP, NIT, or CYC.

Therefore, I recommend revising the section title, "Case studies demonstrate the depth of mechanistic insights afforded by QMS-seq," to something more reflective of this section's speculative nature, perhaps along the lines of L71.

The case studies are partly speculative, and we agree the subheading title was unnecessarily emphatic. We have toned it down to: "QMS-seq provides insight into the mechanisms underpinning resistance", echoing L71 suggested by the Reviewer. We also updated the short paragraph preceding the actual case studies to make their speculative nature explicitly clear (L266).

To further address the Reviewer's comment, we now provide a more detailed and stronger hypothesis about the potential mechanism behind *lpxC* terminator mutations. Namely, immediately downstream of the *lpxC* terminator is the transcriptional unit encoding MutT. This enzyme is responsible for clearing oxidized guanine from the nucleotide pool, and over production of MutT has been shown to reduce susceptibility to ROS-mediated antibiotic killing (PMID 22517853). Thus, increased terminator read-through resulting in increased MutT expression is a likely explanation for the *lpxC* terminator mutations. We added this hypothesis in place of the sentence mentioning colistin-resistance, using language reflecting the speculative nature (L277-284).

Reviewer #2

The authors have performed extra experiments and revised many sections of the paper to account for the reviewer's comments and suggestions. The clarity of the manuscript has been improved and the authors have unveiled new pieces of information through the additional experiments. All my concerns have been adequately addressed.

I have just a very minor comment regarding the new "case studies" they present: for two out of the three examples of synonymous mutations, the mutations are actually located at the very beginning of the genes and in this portion of the gene, it has been

shown that the effect of synonymous mutations is often due to the effect on mRNA secondary structure and access of the ribosome to the Ribosomal Binding Site. In other words, these mutations might affect the level of translation of these genes. This explanation could be added to the manuscript around I290.

We appreciate the suggestion to look deeper into synonymous mutations, which has improved our hypothesis regarding the isoleucine ATC > ATA mutations. We've updated the case study to mention the possibility that changes in mRNA secondary structure near the start codon could influence ribosome binding, as suggested by the Reviewer (L299-301). We also provide more detail regarding the unique decoding mechanism used for ATA (or rather AUA), which could contribute to ribosomal pausing (L297-299)